# Analysis and Prevention of AI-Based Phishing Email Attacks

Chibuike Samuel Eze and Lior Shamir *

Department of Computer Science, Kansas State University, Manhattan, KS 66506, USA; cseze@ksu.edu
* Correspondence: lshamir@mtu.edu

**Abstract:** Phishing email attacks are among the most common and most harmful cybersecurity attacks. With the emergence of generative AI, phishing attacks can be based on emails generated automatically, making it more difficult to detect them. That is, instead of a single email format sent to a large number of recipients, generative AI can be used to send each potential victim a different email, making it more difficult for cybersecurity systems to identify the scam email before it reaches the recipient. Here, we describe a corpus of AI-generated phishing emails. We also use different machine learning tools to test the ability of automatic text analysis to identify AI-generated phishing emails. The results are encouraging, and show that machine learning tools can identify an AI-generated phishing email with high accuracy compared to regular emails or human-generated scam emails. By applying descriptive analytics, the specific differences between AI-generated emails and manually crafted scam emails are profiled and show that AI-generated emails are different in their style from human-generated phishing email scams. Therefore, automatic identification tools can be used as a warning for the user. The paper also describes the corpus of AI-generated phishing emails that are made open to the public and can be used for consequent studies. While the ability of machine learning to detect AI-generated phishing emails is encouraging, AI-generated phishing emails are different from regular phishing emails, and therefore, it is important to train machine learning systems also with AI-generated emails in order to repel future phishing attacks that are powered by generative AI.

**Keywords:** phishing; cybersecurity





## 1. Introduction

A phishing email attack [1–8] is a form of cybersecurity attack in which the attacker contacts a potential victim through an email message that deceives the victim to believe the message was originated from a credited source. Once earning their trust, the victims are led to commit certain actions that they would not have committed if they knew the true identity of the attacker. In most cases, such an attack aims at deceiving the victim to provide private information such as login credentials to certain accounts, but can also include other actions such as transferring money to the attacker. Because the victim of a phishing email attack believes that the message was originated from a credible source, they provide the information willingly. Phishing attacks, therefore, lead to obtaining passwords or other private information that would have been very difficult to obtain otherwise.

Phishing attacks have been in existence long before the emergence of the Internet, and reports on such attacks are dated as early as the first half of the 19th century [9]. An example is the "Letter from Jerusalem" attack that promised the victim a share of a large amount of money, if the victim be willing to provide a relatively small fee that would be used for bribing the prison wardens [9].

With the use of electronic mail, phishing attacks became much more accessible and cheaper to execute. Phishing email attacks are among the most common cybersecurity attacks, and inflict substantial damages on individuals and organizations. Phishing attacks often start by sending emails automatically to a large number of random recipients. The assumption of the attacker is that while most recipients will not respond to the initial email, a small number of the recipients might be deceived into following the instructions of the

attacker. It is, therefore, important for the attacker that the email messages are crafted by using social engineering aspects to optimize the chances of earning the trust of the potential victim [10–13]. To optimize the impact of the attack, it is also important for the attacker to reach as many potential victims as possible, bypassing any defense mechanism that can identify and remove such emails.

## 1.1. Existing Solutions to Phishing Email Attacks

Some early solutions for phishing email attacks were based on simple filtering of information such as URLs, senders, IP addresses, and other elements of the email. These pieces of information can be combined with machine learning to identify phishing emails [14,15]. It has been shown that such attacks can also be detected at the level of the network packets [16]. Mail servers also detect phishing emails as spam, as identical phishing emails are sent to a large number of recipients. Once identified as spam, the email server can delete these identical or nearly identical emails, or move these emails to the spam folder. While this approach can be effective, generative AI can be used to generate a large number of emails such that each one is unique, and therefore, identical or nearly identical emails cannot be detected.

Other approaches are based on natural language processing of the email content [4,17–20]. Such methods in most cases utilize machine learning to classify emails by analyzing their text. Existing solutions are based on Support Vector Machines (SVM) [21], J48 [22], random forest [23], recurrent convolutional neural networks (RCNNs) [24], and other deep neural network architectures [25–31].

These solutions have a certain level of inaccuracy and do not ensure full protection. Therefore, organizations normally train their employees to identify suspected phishing emails and avoid phishing scams [32–34]. Although such training can reduce the likelihood of a person being deceived by a phishing email, attending such training still does not provide a full guarantee that the trained person does not become a phishing scam victim [35].

## 1.2. AI-Generated Phishing Emails

While Generative AI has substantial useful applications, in some cases that technology can be used for malicious purposes. For instance, the ability to create realistic "deep fake" images, videos and audio can be used to disseminate false information [36,37]. Generative AI can also be used for plagiarism or compromising copyrighted art and other human creations. That is conducted by using copyrighted art for training generative AI systems, that consequently generate work heavily based on the copyrighted art it was trained with [38,39]. The ability to create fake identities can be used to spread false information or harass specific targeted victims [40].

Since phishing email attacks are normally implemented by sending phishing emails to a large number of recipients, one of the ways to defend from phishing emails is by identifying a large number of identical emails in the mail server received within a relatively short period of time. While that approach is widely used, identifying identical or very similar emails sent multiple times from external sources can be bypassed by using generative artificial intelligence (AI). Generative AI tools can create different email messages that read as if they were written by a person, although they were generated by a machine and can, therefore, be sent to a large number of people with minimal effort on the side of the attacker. Therefore, the ability to generate a large number of different email messages can be used by an attacker to avoid using the same or similar email message, making it more difficult for spam detection systems to identify repetitive emails sent from an outside source to a large number of recipients.

Here, we study the problem of a phishing email attack that uses generative AI to create a large number of different emails automatically. Such attacks can be more difficult to repel since each email is unique, yet all emails are grammatically correct and use language that would seem convincing to an unsuspecting potential human reader. Since there is no existing email corpus of AI-generated phishing emails, this study proposes a novel email corpus that is unique in the sense that the phishing emails were generated by AI. To study the ability of automatic text analysis to identify AI-generated phishing emails, several different text analysis approaches are tested. This allows us to better understand the ability of automatic text analysis to identify phishing emails generated by AI. Such systems can be used to prevent AI-generated emails that can easily bypass spam protection mechanisms. The AI-generated email corpus is made publicly available and can be used by the community for a variety of purposes related to phishing emails generated by AI.

Section 2 describes the AI-generated phishing email corpus, as well as the other existing email datasets used in this study. Section 3 briefly describes the text analysis methods used to identify AI-generated phishing emails. Section 4 shows the results of applying these text analysis methods to the email datasets, and Section 5 summarizes the conclusions of the study.

## 2. Data

To train and test machine learning models, a sufficiently large dataset of AI-generated phishing emails is required. To create a sufficiently large dataset of AI-generated phishing emails, the API of the *DeepAI* https://deepai.org/ (accessed on 2 May 2024) platform was used. *DeepAI* is enabled by OpenAI, and provides the API that allows the automation of the generation of large numbers of text files. OpenAI is a mature revolutionizing technology that allows transforming the input text queries into output text that corresponds to the input in a comprehensive manner. It is based on the first step of word embedding of the input, followed by a self-attention transformer architecture that can make associations between words based on the data it was trained with. The language model can then be used to produce the text such that each word is generated based on the words that came before it. OpenAI has become a highly common and highly used technology.

The main advantage of *DeepAI* for the purpose of this project is that, unlike other chatbots, it did not refuse to generate phishing emails. Chatbots such as ChatGPT, Copilot, and Character.ai refused to generate phishing emails, and therefore, cannot be used for the study described in this paper. By using the power of *DeepAI*, a dataset of 865 phishing emails was generated. The dataset can be accessed at https://people.cs.ksu.edu/~lshamir/data/ai_phishing/ (accessed on 2 May 2024).

The emails in the dataset are in plain text format. The average length of each email is 545 characters, where the longest email is 1810 characters and the shortest email is 280 characters long. Below is an example of an AI-generated phishing email taken from the dataset:

Dear Valued Customer,

We are contacting you to inform you that your account requires urgent verification to prevent any potential security breaches and unauthorized access. In order to safeguard your account and maintain the security of your personal information, we kindly request you to click on the following link to complete the verification process: www.secureverifylink123.com.

Please note that failure to verify your account within the specified timeframe may result in temporary suspension or limited access to your account. It is essential that you act promptly to avoid any disruption to your account services. If you have any questions or concerns regarding this verification process please contact our customer support team immediately.

Thank you for your prompt attention to this matter and for helping us maintain the security of your account.

Sincerely,
Customer Support Team

The email reads like a typical phishing email, and might lead to the impression that the email was written by a person rather than by a machine. The email expresses a sense of urgency as common in many phishing email attempts, and provides an external link where the recipient of the email is asked to provide their personal information. The email also sets a due date and describes the consequences in case the personal information is not shared. A similar approach is demonstrated by the following AI-generated email.

Dear Grace Adams,

As part of our security measures, all library accounts need to be verified. Please click on the following link to validate your library account: http://bit.ly/89HjeFd

Completing the account validation process within 48 h will ensure uninterrupted access to library resources and services. If you encounter any difficulties or require assistance, please contact our Library Services Team.

Thank you for your swift action in validating your library account.

Best regards,
Library Services Team

The email is certainly different from the first email, but follows the same approach of expressing urgency, requesting personal information through a link, and specifying consequences if the potential victim does not take the requested action. In other emails, the generative AI also added a specific reason for the urgency that supposedly triggered the request for information, as shown in the following email:

> Dear Lily Evans,
>
> Suspicious login attempts have been identified on your online account. To safeguard your account, please verify your information by clicking on the following link: http://bit.ly/4nJhVsW
>
> Completing the verification within 48 h is critical to prevent any potential security risks. If you need assistance or have questions, please reach out to our Online Security Department.
>
> Thank you for your cooperation in securing your online account.
>
> Best regards,
> Online Security Department

While most AI-generated phishing emails were of similar form, some of these emails just requested information in a more positive manner, without specifying consequences for not taking action.

> Hey Piper,
>
> Join our Pet Wellness Workshop for heartwarming tips, expert advice, and caring insights on keeping your furry friends happy, healthy, and loved. Click the link to join the workshop and prioritize your pet's well-being: hxxps://petwellnessworkshop.heartfeltpets123.com
>
> Let's wag tails, share love, and nurture our pet companions together!
>
> Paws up, Animal Lovers Society

To profile the diversity of the emails, we selected 100 random emails from the dataset. A manual analysis of 100 emails selected randomly from the dataset showed that the majority of the phishing emails requested account updates or an urgent verification of the account. These emails made up 48% of the emails. The second largest group of emails were emails reporting on suspicious activities in the account. These emails made up 36% of the emails. Two emails reported that the account was suspended and requested to re-activate it. Seven emails asked to verify financial transactions, four emails were about activities in student accounts, and three emails invited the reader to participate in events.

The dataset of AI-generated phishing emails was tested with several existing email datasets that are commonly used for machine learning tasks. The first is the Enron email dataset [41–44], which is used as a dataset of "regular" email messages. The dataset was collected as part of the Federal Energy Regulatory Commission that examined the collapse of the Enron Corporation. Since released to the public, it has been a common dataset for the development and testing of machine learning methods for numerous tasks such as spam detection [45], categorization into folders [46], network analysis [47], and more. The Enron email dataset is a large email corpus that contains $\sim 5 \cdot 10^5$ emails and is available for download at https://www.cs.cmu.edu/~enron/ (accessed on 6 May 2024).

Another dataset that was used is the dataset of Nigerian scam emails. The Nigerian scam is a form of phishing attempt aimed at deceiving a person to provide financial information or transfer money to the attacker. It became known as an attempt to deceive potential victims to receive a large amount of money as a fee for participating in a government corruption aiming to transfer a very large amount of money outside of Nigeria. An unsuspecting victim might then provide information such as bank account numbers or transfer smaller amounts of money to receive a portion of the much larger amount they believe they can earn.

In addition to these datasets, we also used the Ling-Spam dataset [48,49] as a dataset of manually crafted regular and spam emails. The dataset contains 2412 regular emails and 481 spam emails and can be accessed at https://metatext.io/datasets/ling-spam-dataset (accessed on 6 May 2024).

## 3. Methods

The classification between AI-generated phishing emails and emails written manually is a task that has not been tested before. Therefore, no specific method designed specifically for AI-generated phishing emails has yet been proposed. To test whether AI-generated emails can be identified automatically, several different text analysis methods were tested, each one based on a different text analysis approach. The first method is MAchine Learning for LanguagE Toolkit (MALLET) [50]. MALLET is a mature open-source software tool for topic modeling which is commonly used for automatic document classification. It uses several machine learning algorithms to allow the automatic classification of text documents into classes. MALLET is a general-purpose document classifier, and its ability to classify manually generated spam emails has been demonstrated [51].

MALLET first applies text segmentation to identify the different parts of the document, followed by a step of tokenization and stop word removal. Then, lemmatization is applied, and the parts of speech of each word are determined. That allows to build a dictionary, and the frequency of the dictionary words in each document leads to the feature space. Once the feature space is created, a machine learning algorithm is used to make the classification. MALLET provides several machine learning algorithms from which the user can choose, and these include Winnow, maximum entropy, decision tree, and Naive Bayes. As a classifier based on topic modeling, MALLET can identify differences in the use of words when the email is written by a person compared to emails written by AI.

Another tool that was used was the open-source Universal Data Analysis of Text (UDAT) [52,53]. Unlike MALLET, UDAT does not use topics or keywords identified in the text. Instead, it is based on the analysis of style elements such as the use of parts of speech, repetition of words, punctuation, and more. Text descriptors such as the distribution and pattern repetition of parts of speech are powered by the CoreNLP library [54].

UDAT is a non-parametric method that uses a comprehensive set of descriptors extracted from the text, as fully explained in [53]. In summary, these include the distribution of the lengths of the words, diversity of the words appearing in the text, changes in the frequency of words throughout the document, use of punctuation characters, use of upper case letters, frequency and length of quotations, use of emoticons, distribution of sentence length, use of numbers in the text, distribution of sounds as reflected by the Soundex algorithm, readability index, distribution of parts of speech, repetitive patterns in the use of parts of speech, and analysis of the sentiments and their distribution throughout the document.

Applying that collection of algorithms to the text provides 297 numerical content descriptors that reflect the text. These features are then weighted by using Fisher discriminant scores, and a weighted nearest neighbor classifier is used such that the Fisher discriminant scores are used as weights [53]. UDAT was developed for the purpose of digital humanities [55,56], but can also be used as a general-purpose document classifier [52,53,57].

UDAT can identify differences in style elements as expressed in the text. Therefore, if phishing emails written by AI have certain specific style elements that make them different from emails written by humans, UDAT will be able to identify AI-generated phishing emails even if the distribution of topics and words in the emails does not allow to identify the phishing emails. Therefore, such analysis can be used in addition to the word-based analysis as used by MALLET to provide a more complete defense. One of the advantages of UDAT is that it also provides a specific style element that differentiates between the classes. That is, in addition to classifying between documents, it also points to the specific style elements that make the documents different [53]. That explainable aspect of the analysis

can help understand the specific differences that can be used to identify phishing emails generated using AI.

Another method that was used is based on a deep neural network. The neural network is relatively simple, and based on the common Long Short Term Memory (LSTM) architecture [58]. The first layer is an embedding layer with the maximum number of words set to 20,000, the maximum sequence length set to 50, and 100 embedding dimensions. That layer is followed by LSTM with an input size of 100 and dropout and recurrent dropout both set to 0.2. The activation function is *tanh*, and the recurrent activation function is *sigmoid*. The loss function was the common *Categorical Crossentropy*, and the Adam optimizer was used for training. The last layer is a fully connected layer with *softmax* activation. The code is available at https://gist.github.com/agrawal-rohit/ff2c5defe437abd997fa6c576aa29235 (accessed on 2 May 2024). The neural network was trained with 10 epochs.

The methods were used by randomly allocating 600 text samples from each class for training, and 100 for testing. The analysis was conducted with 10-fold cross-validation. The evaluation of the results was determined by classification accuracy, precision, recall, and F1, which are common ways to evaluate the results of machine learning algorithms. The recall was determined by $\frac{true\,positives}{all\,positives}$, and the precision was determined by $\frac{true\,negatives}{all\,negatives}$. The confusion matrix was also used as a way to profile the results of the machine learning classification and evaluate the performance.

In addition to these methods, an ensemble method was used. The ensemble classifier was made of the deep neural network described above, UDAT, and MALLET with Naive Bayes classifier. The final classification was made by a simple majority rule. That makes a single text classifier made of three different text classifiers such that each one uses a different text analysis approach.

## 4. Results

When using MALLET to classify between the four classes, three different machine learning algorithms were used: Winnow, maximum entropy, and Naive Bayes. The classification accuracy was 99.3% with the Naive Bayes classifier, 99.2% with the Maximum Entropy classifier, and 97% when Winnow was used as a classifier. The statistical significance of the results was determined by applying analysis of variances (ANOVA) using the mean and standard deviation of the classification accuracy. In all cases, the chance to observe such accuracy or better by chance is ($p < 10^{-5}$). Table 1 shows the confusion matrix of the classification when using the Naive Bayes classifier. As the confusion matrix shows, nearly all AI-generated emails were classified correctly. The recall of the detection of AI-generated phishing emails is 0.99, and the precision is 0.98. The F1 is 0.985.

**Table 1.** Confusion matrix of the classification results of the AI-generated emails, the Enron emails, the manually crafted Nigerian phishing scam emails, and Ling-Spam emails when using MALLET and Naive Bayes classifier. The values are in percentage.

|                   | AI-Generated | Enron | Ling-Spam | Nigerian Phishing |
| ----------------- | ------------ | ----- | --------- | ----------------- |
| AI-generated      | 99           | 1     | 0         | 0                 |
| Enron             | 0            | 99    | 1         | 0                 |
| Ling-Spam         | 0            | 1     | 99        | 0                 |
| Nigerian phishing | 1            | 2     | 0         | 97                |

When using *UDAT* for the classification of the four classes, the classification accuracy of the four classes is 98% ($p < 10^{-5}$), with a recall of 1 and precision of 0.97, leading to F1 of 0.985. Table 2 shows the confusion matrix of the classification results. As the table shows, the AI-generated emails were classified with 100% accuracy, and most of the confusion of the classifier was between the classes of emails that were not generated by AI.

**Table 2.** Confusion matrix of the classification results of the AI-generated emails, the Enron emails, the manually crafted Nigerian phishing scam emails, and the Ling-Spam emails. The numbers show percentage of the total number of test samples from each class.

|  | AI-Generated | Enron | Ling-Spam | Nigerian Phishing |
|---|---|---|---|---|
| AI-generated | 100 | 0 | 0 | 0 |
| Enron | 0 | 97 | 0 | 3 |
| Ling-Spam | 0 | 4 | 96 | 0 |
| Nigerian phishing | 0 | 0 | 0 | 100 |

Experiments of two-way classification were also attempted. That included three experiments, such that the first experiment was a two-way classifier between AI-generated emails and Nigerian scam emails, the second experiment was a classifier between AI-generated phishing emails and Enron emails, and the third experiment was a classifier between AI-generated phishing emails and Ling-Spam emails. As expected, in all cases the classification accuracy was 100%.

Because *UDAT* is based on intuitive text descriptors, it can provide the descriptors that are the most informative for the classification. That is conducted through *UDAT* by using the Linear Discriminant Analysis (LDA) scores of the features [59]. The LDA scores are used as weights for the classification [59], but they can also be used to rank the text descriptors by their informativeness [53]. Table 3 shows the most informative numerical text descriptors that have the highest LDA score.

**Table 3.** Means of the features with high LDA scores and the mean of each feature in each email corpus. The *p*-values are based on ANOVA analysis with four groups.

| Text Descriptor | AI-Generated | Enron | Ling-Spam | Nigerian Phishing | $p$ (ANOVA) |
|---|---|---|---|---|---|
| Pronoun frequency | $0.1014 \pm 0.0014$ | $0.06 \pm 0.001$ | $0.06 \pm 0.0005$ | $0.077 \pm 0.001$ | $<10^{-5}$ |
| Verb frequency | $0.1066 \pm 0.001$ | $0.052 \pm 0.0007$ | $0.0319 \pm 0.0005$ | $0.044093 \pm 0.001$ | $<10^{-5}$ |
| Word length mean | $5.672 \pm 0.014$ | $4.82 \pm 0.016489$ | $4.98 \pm 0.012488$ | $4.76 \pm 0.034008$ | $<10^{-5}$ |
| POS diversity | $0.253 \pm 0.004$ | $0.103 \pm 0.001$ | $0.108 \pm 0.0026$ | $0.1 \pm 0.005$ | $<10^{-5}$ |
| Noun frequency | $0.17 \pm 0.002$ | $0.188 \pm 0.002$ | $0.277 \pm 0.002$ | $0.211 \pm 0.04$ | $<10^{-5}$ |
| Sentiment mean | $1.68 \pm 0.013$ | $1.3 \pm 0.01$ | $1.506 \pm 0.005$ | $1.26 \pm 0.016$ | $<10^{-5}$ |
| Coleman–Liau index | $13.89 \pm 0.095$ | $10.2 \pm 0.097$ | $11.37 \pm 0.078$ | $10.86 \pm 0.19$ | $<10^{-5}$ |
| Cardinal number frequency | $0.0037 \pm 0.0002$ | $0.0371 \pm 0.001$ | $0.0447 \pm 0.001$ | $0.029837 \pm 0.002$ | $<10^{-5}$ |
| Past tense verb frequency | $0.00048 \pm 0.0001$ | $0.01357 \pm 0.000414$ | $0.0062 \pm 0.0003$ | $0.0138 \pm 0.0004$ | $<10^{-5}$ |
| Lemma diversity | $0.719 \pm 0.0037$ | $0.57 \pm 0.003$ | $0.568 \pm 0.004$ | $0.526 \pm 0.006$ | $<10^{-5}$ |
| Sentence length mean | $8.815 \pm 0.086$ | $16.045 \pm 0.285682$ | $16.7 \pm 0.200506$ | $16.58 \pm 10.981747$ | $<10^{-5}$ |

As the table shows, there are several statistically significant differences between AI-generated phishing emails and other emails. For instance, AI-generated emails have many more verbs and pronouns compared to emails written manually. On the other hand, AI uses fewer cardinal numbers and fewer past tense verbs. These differences allow us to classify between an AI-generated phishing email and other emails.

Also, the average length of the words in AI-generated phishing emails is significantly longer than words in emails written manually. The average word in an AI-generated email is ~5.7 characters, while the average word length in the other email classes ranges between 4.76 and 5 characters. That observation can be potentially explained by the labor involved in typing longer words manually, or by the larger vocabulary required to be familiar with more long words and their meaning. Naturally, AI does not have labor or effort considerations when generating an email message. Also, artificial intelligence is not limited by its memory capacity and, therefore, has access to a larger vocabulary compared to a person. The shortest words were used in the Nigerian phishing emails, which is possibly due to the fact that many of these scams originated in countries where the formal language is not English.

The analysis also shows that AI-generated emails are more diverse in the words they use, meaning that they do not tend to repeat the same words compared to manually written emails. As described in [53], the lemma diversity is determined by the number of different lemmata used in the text divided by the total number of lemmata in the text. When the same lemma is used more times in the same text, the lemma diversity becomes smaller. Manually written emails tend to repeat the same lemmata, while AI-generated emails tend to avoid repeating the same lemmata and use a more diverse vocabulary.

Another difference is that AI-generated emails express more positive sentiments. The sentiments of the text are estimated by applying the sentiment analysis of CoreNLP [54] to each sentence, annotating each sentence in the text with a sentiment score between 0 and 4, where 0 means very negative and 4 means very positive. The analysis shows that AI-generated emails express more positive sentiments, while the Nigerian phishing scam emails express the most negative sentiments.

The artificial neural network described in Section 3 was used to classify between the AI-generated phishing emails and each one of the other classes. Like with the other methods, the classification accuracy is high. When classifying between AI-generated phishing emails and Nigerian phishing fraud the classification accuracy is 100% ($p < 10^{-5}$). When classifying between the AI-generated phishing emails and emails from the Enron corpus the classification accuracy is 99%, with recall of 0.99, precision of 0.98, and F1 of 0.985. The classification accuracy between AI-generated phishing emails and spam emails from the Ling-Spam corpus is also 99%. In all cases the probability of observing such classification accuracy or stronger by chance is ($p < 10^{-5}$). The experiments were conducted with 10 epochs, but maximum accuracy was reached already after five epochs.

In summary, all algorithms that were tested showed good results in identifying automatically generated phishing emails. Figure 1 shows the classification accuracy of the different algorithms that were tested. As the figure shows, MALLET with the Bayes Network classifier provided the best performance, but the differences between the algorithms are minor. The method using a simple majority voting provided an accuracy of 99.5%, which is a minor improvement over using MALLET alone.

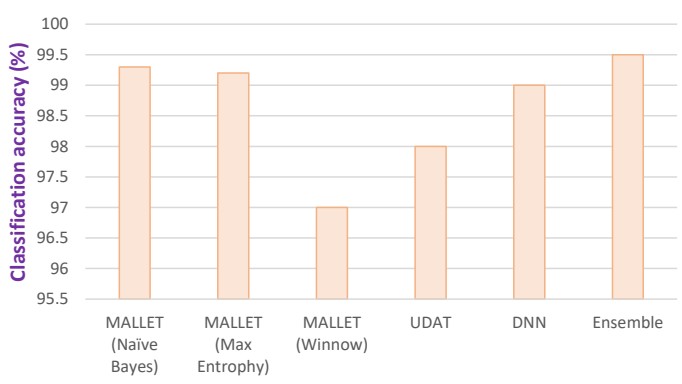

**Figure 1.** Classification accuracy when using the different text analysis algorithms.

## 5. Conclusions

Phishing email scams are among the most common and most harmful cybersecurity attacks. Despite the substantial damage they inflict on individuals and organizations, there is still no perfect solution to repel phishing attacks.

The availability of generative artificial intelligence can be used to make phishing email attacks more sophisticated by generating a large set of unique phishing emails, such that each victim obtains a different email or even several different emails, each of them unique. Because phishing email attacks are based on sending a large number of emails to a large number of potential victims, using a unique email can help the attacker bypass systems

that identify identical emails and remove them from the system or copy them into the spam folder.

Here, we describe a corpus of phishing emails generated automatically by generative artificial intelligence and test the ability of different types of text classifiers to identify between AI-generated phishing emails and emails written by humans. The purpose of this work is not necessarily to propose a new text analysis algorithm, but to make the corpus available, and test the efficacy of several different existing text analysis paradigms for their ability to identify AI-generated phishing emails. The results can be used to understand how effective these systems are, and also provide baselines for comparison when new algorithms are proposed.

Because generative AI is trained with manually generated data, it could be assumed that AI-generated phishing emails are similar to manually prepared phishing emails. However, as the analysis shows, there are identifiable differences between human-generated and AI-generated emails, and in fact, machine learning can identify AI-generated phishing emails with high accuracy. All the results are statistically significant. AI-generated phishing emails are different from regular emails, but also from manually-generated phishing scam emails. Some of the specific differences can be profiled, and show that the basic writing style used by generative AI is not identical to emails written manually, and therefore, makes these emails different.

These results are encouraging, and show that neural networks, topic modeling and the analysis of the style elements can identify between phishing emails generated by AI and human-written emails. Still, to allow such identification of AI-generated phishing emails the machine learning systems need to be trained on AI-generated emails, as these emails are different from manually-written emails. The analysis also showed specific differences between emails generated by AI and regular emails. The corpus of AI-generated emails prepared for this study is available publicly and can be used to develop new methods related to phishing scams powered by generative artificial intelligence.

One of the observations made in this study is that despite the differences between the different text classification approaches, all approaches were able to identify the AI-generated phishing emails with high accuracy. An effective solution would, therefore, be based on several algorithms that work in concert rather than a single approach. That can ensure that if the attacker crafts their AI-generated emails to overcome one approach of text classifiers, the other text classifiers can still detect the phishing attack. That can provide a form of defense that is more difficult to penetrate, as the attacker needs to adjust their AI to generate emails that can deceive several different approaches of detection rather than merely one.

Machine learning tools that identify AI-generated phishing emails might not be sufficiently robust to delete suspected emails automatically, as even high detection accuracy still leads to the deletion of valid emails. Even if such deletions are rare, automatic deletion of emails or moving them into a different folder might make such systems more difficult to deploy. Also, the analysis was conducted by comparing AI-generated emails to several known and widely used email datasets. Because it is impractical to represent all possible emails, it is possible that some types of emails not studied here might be able to deceive the machine learning used here. In such cases new machine learning will be needed, ideally trained on the same emails that deceived it to avoid future similar emails to penetrate the system. That, however, might be difficult to do before the types of attacks committed are executed, or the emails that the system fails to identify are known.

The same also applies to the AI-generated phishing emails. The corpus of phishing emails used here is based on emails of different phishing approaches as described in Section 2. That model might be biased, and might not necessarily represent all possible future AI-generated phishing email attacks. An attacker might generate phishing emails in different forms, designed for a specific purpose, or modify the emails after they are generated by the AI system. That might also limit the ability of a detection system to identify AI-generated emails in a given attack that has not been used in the past.

But despite their limitations, such machine learning systems can still be used to provide a warning that a certain email message is suspected to be part of a phishing email attack. Such warnings can be given to the user or the system administrator who can examine the suspicious emails as they are received. Since advanced generative AI has been becoming more powerful as well as more accessible, such attacks powered by generative AI are expected. Fortunately, the results shown here suggest that machine-learning tools can identify such emails in high accuracy. It is, therefore, important to integrate such a system into email to repel future phishing scams that make use of the power of generative AI.

To the best of our knowledge, the email corpus described in this paper is the first corpus of AI-generated phishing emails. The availability of the corpus will enable new studies on tasks related to AI-generated phishing email attacks. The results reported here can be used as a baseline, but it is possible that the corpus introduced in this study will be used in a variety of ways and in combination with different kinds of datasets to further advance research related to AI-generated phishing emails.

**Author Contributions:** Conceptualization, L.S.; Methodology, L.S. and C.S.E.; Software, C.S.E.; Validation, L.S. and C.S.E; Data curation, C.S.E. and L.S.; Writing—original draft, L.S. All authors have read and agreed to the published version of the manuscript.

**Funding:** This research was funded by NSF grant number 2148878.

**Data Availability Statement:** The data underlying this article are public datasets. The only dataset created for this research is available at https://people.cs.ksu.edu/~lshamir/data/ai_phishing/ (accessed on 2 May 2024).

**Conflicts of Interest:** The authors declare no conflicts of interest.

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
