# Peer review of "Analysis and Prevention of AI-Based Phishing Email Attacks"

_electronics, doi:10.3390/electronics13101839_

Round 1

Reviewer 1 Report

Comments and Suggestions for Authors

In this paper, some results of experiments of classifying AI-Base phishing email are presented. Only one engine prepared by OpenAI is used to generate Ai-Base phishing email, and some exciting methods are applied to the classification, and on these points, this paper is not novel.

However, the explanation of  results of the experiments is good, and the readers will feel interesting. 

As an improvements of this paper, 1, Add the another method to generate AI-base email,  2. Add another original method for classification for example ensemble of the methods used in this paper. 

Author Response

In this paper, some results of experiments of classifying AI-Base phishing email are presented. Only one engine prepared by OpenAI is used to generate Ai-Base phishing email, and some exciting methods are applied to the classification, and on these points, this paper is not novel.

--Author response: Thank you for the time you spent reading and commenting on the manuscript. We would like to note that while the paper does not proposes a new algorithm, this is the first email corpus of AI-generated phishing emails, and no such corpus exists. It is also the first attempt to identify phishing emails generated by AI. AI-generated phishing emails can be used to penetrate through defense systems since each email is different, and therefore comparison of the emails to identify a large number of identical emails in the mail server becomes impossible. Attackers can therefore use AI-generate emails to attack. That problem has not been studied before. The introduction section has been changed the better reflect that, and a special sub-section has been added to the Introduction to explain that.

However, the explanation of  results of the experiments is good, and the readers will feel interesting. 

--Author response: Thank you for the kind words. The replies for the other comments are below. Changes made to the manuscript are highlighted in bold font.

As an improvements of this paper, 1, Add the another method to generate AI-base email,  2. Add another original method for classification for example ensemble of the methods used in this paper. 

--Author response: An ensemble of the methods has been added to the Results section, and is also described briefly in the revised Methods section. Because the classification accuracy is high, the ensemble does not provide a substantial improvement. In fact, because the classification accuracy is high by all methods tested in the paper, no method will provide a substantial improvement. But the ensemble of the other method has been added. As mentioned above, the innovation of the paper is not necessarily algorithmic, but the new email corpus, the problem of identifying AI-generated emails, and the observations made by applying different approaches to the problem.

The other comment is more tricky. The problem is that most chatbots refused to generate a phishing email. Some chatbots like ChatGPT and character.ai say they cannot engage in such activity, while Copilot just refuses to give an answer. We tested numerous chatbots until we found the one that “agreed” to generate phishing emails. The resistance of many chatbots to generate phishing emails has been added to Section 2 in the revised version of the paper.

Reviewer 2 Report

Comments and Suggestions for Authors

The manuscript addresses a timely and relevant issue in cybersecurity - the use of generative AI to craft phishing emails. The novelty lies in its focus on AI-generated phishing emails, a relatively less explored area compared to traditional phishing strategies. The introduction of a corpus of AI-generated phishing emails for public use is also a significant contribution that could stimulate further research.

Suggestions for Improvement:

- Enhance the methodology section with more details on the machine learning models, including hyperparameters, training details, and model selection criteria.

-Include a more detailed statistical analysis of the model's performance, addressing both the effectiveness and potential biases.

- Expand the corpus description to include information on diversity and representativeness.

- The manuscript would benefit significantly from the inclusion of a specific framework, model, or algorithm developed by the authors. This addition would not only enhance the originality but also the applicability of the research.

- While the use of machine learning tools is mentioned, there is a lack of depth in the discussion about how each algorithm operates and its suitability for the problem at hand.

- The manuscript currently lacks graphical representations of the results, which could aid in better understanding and dissemination of the findings.

- There is an insufficient comparison with existing literature, which is crucial for highlighting the contributions and positioning of the manuscript within the current research landscape.

- The distinction between AI-generated and manually crafted emails, based on style differences, is particularly insightful. However, quantitative data such as precision, recall, and accuracy metrics would enhance the clarity and impact of the results presented.

- the manuscript would benefit from a more detailed description of the algorithms used, feature extraction techniques, and the composition of the training and test datasets. A more thorough explanation of the evaluation metrics and validation techniques would also be beneficial.

- The current manuscript does not extensively explore different scenarios or test cases, which could limit the generalizability of the conclusions.

- To reach publication standards, the manuscript needs to ensure a polished presentation style and flawless grammar.

- The manuscript should reflect the latest developments and research in the field to be considered up-to-date and relevant.

References Suggestion:
Atawneh, S., & Aljehani, H. Phishing Email Detection Model Using Deep Learning. Electronics, 12(20), 4261. https://doi.org/10.3390/electronics12204261

Altwaijry, N., Alotaibi, R., & Alakeel, F. Advancing Phishing Email Detection: A Comparative Study of Deep Learning Models. Sensors, 24(7), 2077. https://doi.org/10.3390/s24072077

Haytham Tarek Mohammed Fetooh, M. M. EL-GAYAR and A. Aboelfetouh, “Detection Technique and Mitigation Against a Phishing Attack” International Journal of Advanced Computer Science and Applications(IJACSA), 12(9), 2021. http://dx.doi.org/10.14569/IJACSA.2021.0120922

S. Salloum, T. Gaber, S. Vadera and K. Shaalan, "A Systematic Literature Review on Phishing Email Detection Using Natural Language Processing Techniques," in IEEE Access, vol. 10, pp. 65703-65727, 2022, doi: 10.1109/ACCESS.2022.3183083.

Comments on the Quality of English Language

Moderate editing of English language required

Author Response

The manuscript addresses a timely and relevant issue in cybersecurity - the use of generative AI to craft phishing emails. The novelty lies in its focus on AI-generated phishing emails, a relatively less explored area compared to traditional phishing strategies. The introduction of a corpus of AI-generated phishing emails for public use is also a significant contribution that could stimulate further research.

--Author response: Thank you for the time you put in reading and commenting on the manuscript, and for the kind words. We were given a relatively short time to respond to all comments, but we believe we addressed all of them. The responses for the comments and the corresponding changed made to the manuscript are specified below each comment. Changes made to the manuscript are highlighted in bold font.

Suggestions for Improvement:

- Enhance the methodology section with more details on the machine learning models, including hyperparameters, training details, and model selection criteria.

--Author response: Thank you for the comments: A lot of information about the neural network has been added, including the drop out, the activation functions, the loss function, the optimizer, the number of dimensions in the LSTM layer, and other parameters such as the maximum number of words and the sequence length. The code is open, but the description now includes everything needed to reproduce the neural network even without the code.

-Include a more detailed statistical analysis of the model's performance, addressing both the effectiveness and potential biases.

--Author response: It is indeed common that machine learning results are provided without statistical inference, and that should not be the case. We added statistical analysis of the results, mostly using ANOVA. The classification accuracy is so high that the probability of these results to occur by chance is extremely low. Table 3 has also been enhanced with the ANOVA p values of the different features. Also, a short discussion about possible biases has been added to the conclusion section. Biases can happen when the emails are not represented by the emails in the training set, and that has now been added to the conclusion section.

- Expand the corpus description to include information on diversity and representativeness.

--Author response: That has been added to the data section. What we did was to observe manually 100 emails and divide them into several broad groups. Most emails are requests to activate the account, and the second largest group is the emails that report on suspicious activities. But there are other smaller groups of emails too. The descriptions and exact numbers have been added to the Data section in the revised version of the manuscript.

- The manuscript would benefit significantly from the inclusion of a specific framework, model, or algorithm developed by the authors. This addition would not only enhance the originality but also the applicability of the research.

--Author response: The UDAT algorithm to analyze text was developed by the authors, but it is described in a separate paper that was published in 2021. The purpose of this paper was to introduce the first AI-generated phishing email corpus, and to use several known approaches to identify them. By making the data public, we enable the development of new algorithms that will surely be proposed in the future. But the topic of the paper is the corpus and the response of AI-generated emails to existing approaches. Also, the classification accuracy is very high already. The results shown in the paper can also be used as baselines when new algorithms are proposed. These notes have been added to the Conclusion section of the paper. 

- While the use of machine learning tools is mentioned, there is a lack of depth in the discussion about how each algorithm operates and its suitability for the problem at hand.

--Author response: That has been added to the revised Section 3. The task of identifying AI-generated phishing emails is a new task that has not been studied before. So in the absence of previous results in the literature, different approaches had to be used to test and find out what the best approach is. More information about each algorithm has been added to the revised paper. Obviously, each method was described in a full paper dedicated to it, so full description cannot be provided (the paper is already nearly 8000 word long), but more information about each method has been added. More importantly, the rationale of using each approach is now better described. The MALLET approach is based on topic modelling, and can identify if AI chooses a different set of words compared to manual analysis. The UDAT approach is based on style elements, and can provide protection by identifying the style differences between humans and AI. To provide full protection, all approaches should be used so that if one approach is compromised by he attacker, the others can still provide defense. That has been added to the conclusion section.

- The manuscript currently lacks graphical representations of the results, which could aid in better understanding and dissemination of the findings.

--Author response: A figure (Figure 1) has been added to the Results section as suggested.

- There is an insufficient comparison with existing literature, which is crucial for highlighting the contributions and positioning of the manuscript within the current research landscape.

--Author response: That is a good point. This paper is the first to propose a corpus of AI-generated emails. That limits the ability to compare to previous literature results, because until now no such email corpus existed. The other datasets used in the paper were used numerous times in the past, and that is because they were made available to the community. That is normally the case, where datasets become available, and right after the methodology is applied to the data, leading to new literature. That point is now made in both the conclusion section and discussion section, explaining where the paper stands within the existing literature. The results reported here can be used as baseline, but it is possible that the corpus introduced in this study will be used in a variety of ways and in combination with different kind of datasets to further advance research related to AI-generated phishing emails.

- The distinction between AI-generated and manually crafted emails, based on style differences, is particularly insightful. However, quantitative data such as precision, recall, and accuracy metrics would enhance the clarity and impact of the results presented.

--Author response: The precision and recall are derived from the confusion matrices, but to make it more convenient to the reader the recall and precision for the different experiments are now specified in the paper. F1 has also been added.

- the manuscript would benefit from a more detailed description of the algorithms used, feature extraction techniques, and the composition of the training and test datasets. A more thorough explanation of the evaluation metrics and validation techniques would also be beneficial.

--Author response: Additional description has been added for the neural network, UDAT, and MALLET. UDAT and MALLET are fairly complex and it is not possible to fully describe them in a single section. Each one of these algorithms is described in a long paper, and the papers are referenced in the manuscript. But the revised version now provides much more information about all the algorithms used.in the experiments. That includes MALLET, UDAT, and the neural network.   

- The current manuscript does not extensively explore different scenarios or test cases, which could limit the generalizability of the conclusions.

--Author response: It is correct that the analysis is focused on standard performance evaluation metrics. It is not clear what the other test cases are, but the paper follows basic and very common practices. But the Conclusion section now mentions that the availability of the corpus enables many other experiments, that will probably be made in the future by members of the scientific community. Just like Ling-Spam an Enron were used in numerous papers just because they were available, it is very reasonable to assume that many more experiments will be made with the dataset released here.   

- To reach publication standards, the manuscript needs to ensure a polished presentation style and flawless grammar.

--Author response: We identified grammar errors and typos in the manuscript, and these have been corrected. The manuscript is now believed to have no grammar errors or misspelled words.

- The manuscript should reflect the latest developments and research in the field to be considered up-to-date and relevant.

References Suggestion:
Atawneh, S., & Aljehani, H. Phishing Email Detection Model Using Deep Learning. Electronics, 12(20), 4261. https://doi.org/10.3390/electronics12204261

Altwaijry, N., Alotaibi, R., & Alakeel, F. Advancing Phishing Email Detection: A Comparative Study of Deep Learning Models. Sensors, 24(7), 2077. https://doi.org/10.3390/s24072077

Haytham Tarek Mohammed Fetooh, M. M. EL-GAYAR and A. Aboelfetouh, “Detection Technique and Mitigation Against a Phishing Attack” International Journal of Advanced Computer Science and Applications(IJACSA), 12(9), 2021. http://dx.doi.org/10.14569/IJACSA.2021.0120922

S. Salloum, T. Gaber, S. Vadera and K. Shaalan, "A Systematic Literature Review on Phishing Email Detection Using Natural Language Processing Techniques," in IEEE Access, vol. 10, pp. 65703-65727, 2022, doi: 10.1109/ACCESS.2022.3183083.

--Author response: Thank you for the suggestion. The last reference was already in the original version of the manuscript, but we added the other three references to the paper.

Reviewer 3 Report

Comments and Suggestions for Authors

This study looks at the rise of phishing attacks caused by generative AI and the development of machine learning tools to detect these threats. Through presenting a corpus of AI-generated phishing emails and test several machine learning models to identify these emails, it concludes that AI-generated emails must be included in training datasets so that machine learning systems can adapt to evolving phishing tactics.

1. Introduction

a. The Introduction section could be significantly improved in terms of clarity and structure. The initial paragraphs repeatedly mention phishing attacks, which could be consolidated and presented more effectively. Furthermore, the section lacks a clear statement of the specific problems being addressed and the corresponding research objectives. It’s difficult to understand the relationship between the background information, the problem statements, and the study's objectives. A more structured approach, with well-defined problem statements followed by clearly stated research objectives, would significantly improve the introduction's coherence and flow. 

b. I agree that the author clearly introduces the issue of AI-driven phishing attacks. It could benefit from a more in-depth discussion of the specific characteristics that distinguish AI-generated phishing from conventional methods. For example, describing how generative AI is currently being used maliciously, which provides readers with a clearer picture of the threat landscape in this problem domain.

c. Please also describe the structure of the manuscript.

2. Dataset

a. A primary concern is the lack of detailed information on the parameters and configurations used in generating the phishing emails, which raises questions about the representativeness of these emails to real-world scenarios. Without a clear understanding of how the AI models were tuned, including the training data used, the diversity of email styles, and the inclusion of various phishing techniques, it's difficult to assess whether the dataset comprehensively covers the spectrum of AI-driven phishing strategies.  

b. The author should provide a comprehensive description of how the AI-generated phishing emails were created. This includes detailing the algorithms or models used, the parameters set for generation, the training data sources, and any preprocessing steps involved. Such documentation will enhance the transparency and reproducibility of the research.

3. Method

a. The Method section mentions using 600 samples for training and 100 for testing, but it does not specify whether the data split was stratified based on the type of phishing emails or completely random. This detail is critical because an unbalanced or non-representative split could result in overfitting or underfitting, reducing the generalizability of the results. 

b. Although the use of MALLET, UDAT, and LSTM is justified, the author does not discuss why these particular models were chosen over others, nor does it detail the configuration settings of each model (like parameters for the algorithms used in MALLET or the architecture specifics of the LSTM model).

c. The primary metric used here is classification accuracy, which is supplemented by a confusion matrix. However, there still are other important metrics not mentioned, such as precision, recall, and the F1-score. The author should also consider them or justify the metrics they used. 

d. The method section should be more structured. Suggest organizing the method around a more detailed description of the data handling processes, model configurations, and validation procedures. Each part should be described in details. 

4. Limitation

The potential limitations of the study are not discussed, such as susceptibility of AI models to new phishing tactics not represented in the dataset, overfitting, or bias in AI-generated training samples. The author should add a limitations subsection discussing these aspects. Consider the impact of model choice, data diversity, and training strategies on the overall robustness and reliability of the phishing detection.

Comments on the Quality of English Language

As stated in review comment.

Author Response

This study looks at the rise of phishing attacks caused by generative AI and the development of machine learning tools to detect these threats. Through presenting a corpus of AI-generated phishing emails and test several machine learning models to identify these emails, it concludes that AI-generated emails must be included in training datasets so that machine learning systems can adapt to evolving phishing tactics.

--Author response: Thank you for the time you spent reading and reviewing our manuscript and for the insightful comments. The replies to the comments with the description of the changes made to the manuscript are specified below each of the specific comments. Changes made to the manuscript are highlighted in bold font.

1. Introduction
a. The Introduction section could be significantly improved in terms of clarity and structure. The initial paragraphs repeatedly mention phishing attacks, which could be consolidated and presented more effectively. Furthermore, the section lacks a clear statement of the specific problems being addressed and the corresponding research objectives. It’s difficult to understand the relationship between the background information, the problem statements, and the study's objectives. A more structured approach, with well-defined problem statements followed by clearly stated research objectives, would significantly improve the introduction's coherence and flow. 

--Author response: Thank you for the comments. We agree that the Introduction section in the original paper was imperfect. The Introduction section has been changed significantly. It was separated into three parts, with subsections separating between them, each sub-section discusses a different topic to make the introduction organized better. Some information has been added, and especially the problem statement is stated in the sub-section 1.2 that is pretty much dedicated to the problem statement.

b. I agree that the author clearly introduces the issue of AI-driven phishing attacks. It could benefit from a more in-depth discussion of the specific characteristics that distinguish AI-generated phishing from conventional methods. For example, describing how generative AI is currently being used maliciously, which provides readers with a clearer picture of the threat landscape in this problem domain.

--Author response: Generative AI can indeed become a problem not just for phishing emails, but also to a broad range of topics. That has been added to the beginning of the new sub-section 1.2, with a few examples of bad uses of generative AI and references to previous papers. 

c. Please also describe the structure of the manuscript.

--Author response: That has now been added to the end of Section 1.

2. Dataset
a. A primary concern is the lack of detailed information on the parameters and configurations used in generating the phishing emails, which raises questions about the representativeness of these emails to real-world scenarios. Without a clear understanding of how the AI models were tuned, including the training data used, the diversity of email styles, and the inclusion of various phishing techniques, it's difficult to assess whether the dataset comprehensively covers the spectrum of AI-driven phishing strategies.  

--Author response: That is indeed one of the weaknesses of generative AI. Generative AI is known to be unexplainable. Even the developers of ChatGPT claim that they do not know how ChatGPT works and cannot explain it (e.g., https://www.vox.com/unexplainable/2023/7/15/23793840/chat-gpt-ai-science-mystery-unexplainable-podcast ). They know that it works based on empirical observations, but cannot explain how based on the algorithms or parameters. Section 2 has been expanded substantially with analysis of the types of emails and their diversity. For instance, we read 100 random emails and sorted them by the phishing techniques used to show how the data are distributed. Other information has also been added. Since the dataset is open people can read the emails and see what they read like, and several examples have been added to the paper. Empirical analysis is probably the only way to understand the email corpus, and we therefore added substantial information about that to the revised Section 2.

b. The author should provide a comprehensive description of how the AI-generated phishing emails were created. This includes detailing the algorithms or models used, the parameters set for generation, the training data sources, and any preprocessing steps involved. Such documentation will enhance the transparency and reproducibility of the research.

--Author response: As specified above, OpenAI is very difficult to understand, and even its creators are not able to understand how it work. It is a very complex system. We added a brief description of OpenAI in the beginning of Section 2, but it is a very complex system and it is not practical to describe it in full details as part of a paper that uses the technology but is focused on something else. Because of the substantial computing resources OpenAI requires to train, it is also not practical for a person to reproduce it from scratch. The reproducibility of the research is by using DeepAI to generate the phishing emails. That should be relatively easy for a reader to do if they want to re-generate the dataset.

3. Method
a. The Method section mentions using 600 samples for training and 100 for testing, but it does not specify whether the data split was stratified based on the type of phishing emails or completely random. This detail is critical because an unbalanced or non-representative split could result in overfitting or underfitting, reducing the generalizability of the results. 

--Author response: The allocation to training and testing was done randomly, and that has now been added to Section 3. Cross-validation is also used to avoid over-fitting as much as possible. The problem might not necessarily be overfitting, but different unexpected uses of AI-generated phishing emails, including emails modified after they were generated to adjust them for the attack. That has been added to Section 5.

b. Although the use of MALLET, UDAT, and LSTM is justified, the author does not discuss why these particular models were chosen over others, nor does it detail the configuration settings of each model (like parameters for the algorithms used in MALLET or the architecture specifics of the LSTM model).

--Author response: Yes. A lot of information has been added about all three methods. Each one is a dedicated paper so the methods cannot be described in all details in one paper (that is basically focused on something else, and is close to 8000 words in length already), but much more information has been added. The full information is available in the cited papers, but this paper now includes a much more detailed descriptions of the methods in Section 3.

c. The primary metric used here is classification accuracy, which is supplemented by a confusion matrix. However, there still are other important metrics not mentioned, such as precision, recall, and the F1-score. The author should also consider them or justify the metrics they used. 

--Author response: That is correct. Precision and recall have been added to all the experiments, as well as F1 scores. ANOVA and p-values for the classification accuracies were also added, as well as the p values for the features studied in Table 3.

d. The method section should be more structured. Suggest organizing the method around a more detailed description of the data handling processes, model configurations, and validation procedures. Each part should be described in details. 

--Author response: A lot more information about the methods has been added to the revised Section 3. Each one of the methods has its own paper (or more than one in the case of MALLET) that describes it, and these papers are fairly long. But a summary of each method has been added. The paper is nearly 8,000 words long, so full detailed description of these methods is not practical, and the full details are available in the cited papers. But more information about each method has been added, an the methods are more clear in the revised version. 

4. Limitation
The potential limitations of the study are not discussed, such as susceptibility of AI models to new phishing tactics not represented in the dataset, overfitting, or bias in AI-generated training samples. The author should add a limitations subsection discussing these aspects. Consider the impact of model choice, data diversity, and training strategies on the overall robustness and reliability of the phishing detection.

--Author response: That is an important point. A lot of information about biases and limitations has been added to the revised Section 5. That includes biases in the data, and the inability to predict the types of data in both the manually prepared “regular” emails and the AI-generated emails. Emails can also be modified an adjusted automatically after being generated by the AI which add further complications. That has been added to the Conclusion section.

Round 2

Reviewer 2 Report

Comments and Suggestions for Authors

The authors have made substantial improvements to the manuscript. In light of these revisions, I believe the manuscript has been significantly enhanced and I am satisfied with the current state of the work.

Comments on the Quality of English Language

 Minor editing of English language required

Reviewer 3 Report

Comments and Suggestions for Authors

The review comments have been addressed. I'm happy with this revision. 

Comments on the Quality of English Language

Please take care of proofreading before the submission.